# Abalone adhesion: The role of various adhesion forces and their proportion to total adhesion force

Peng Xi[1]*, Shaobo Ye[1], Qian Cong[2]

**1** College of Agricultural Engineering, Shanxi Agricultural University, Jinzhong, China, **2** Key Laboratory of Bionic Engineering, Ministry of Education, College of Biological and Agricultural Engineering, Jilin University, Changchun, China

* yydljxp2000@126.com

**Data Availability Statement:** All relevant data are within the paper.

**Funding:** This project was supported by the National Natural Science Foundation of China (Grant No. 51775234), Scientific and Technological

## Abstract

Adhesion is the basic ability of many kinds of animals in nature, which ensures the survival and reproduction of animal populations. The aquatic abalone has a strong adhesion capacity. In this study, we observed the microscopic morphology of abalone abdominal foot surface, and found that the surface was covered with a large number of fibers. Then five types of force measuring plates were designed and processed for the adhesion test of abalone abdominal foot. According to the test results, the composition of abalone abdominal foot adhesion force was analyzed and the proportion of various adhesion force to the total adhesion force of abalone abdominal foot was calculated. Among them, the vacuum adhesion force accounts for more than half of the total adhesion force of abalone abdominal foot, and its proportion is more than 60%. Van der Waals force also plays an important role, and its proportion is more than 20%. The proportion of capillary force is very small, which is only about 1%. Its main role is to form a liquid film to prevent the gas from flowing into the sucker. The vacuum adhesion of abalone abdominal foot can be further divided into the whole adhesion of abdominal foot, the local adhesion of abdominal foot and the frictional equivalent vacuum adhesion. And the whole adhesion of abdominal foot is basically equivalent to the local adhesion of abdominal foot. This study quantifies the proportion of various adhesion forces to the total adhesion force of the abdominal foot, which provides a reference for the further study of other adhesive creatures and the design of bionic underwater adhesion devices.

## Introduction

The environment varies from region to region on earth. In order to better adapt to the nature environment and facilitate the survival and reproduction of themselves and the populations, creatures have formed unique and superior survival skills after a long period of continuous evolution. Adhesion is the basic ability of many animals in nature [1]. Through adhesion, animals can be firmly attached to the corresponding surface to facilitate basic survival activities such as crawling, predation, and reproduction [2]. Many mammals on land have the ability to

Innovation Programs of Higher Education Institutions in Shanxi (2021L162), the doctoral scientific research launch project of Shanxi Agricultural University (2021BQ35) and the Scientific Research Project of Reward Fund for Doctoral Graduates in Shanxi Province (SXBYKY2021076) The funders had no role in study design, data collection and analysis, decision to publish, or preparation of the manuscript.

**Competing interests:** The authors have declared that no competing interests exist.

climb trees, and they climb through the adhesion force formed by powerful claws embedded in the inside of the tree [3]. The gecko adheres by van der Waals force formed between the toe and the contact surface to ensure its free crawling on the wall and roof [4–7]. Adhesion is not only common in terrestrial animals, but also in some aquatic animals [8, 9]. The octopus can hold the object firmly through the sucker on its wrist [10–15]. Leeches can be blood-sucking and move inchworms through suckers [16–18]. Through the sucker structure of the head, the remora can be firmly attached to the surface of the shark and the bottom of the ship, thus moving with them [19–22]. The clingfish can be attached to the rock surface through the sucker of the abdomen to prevent it from being washed away by the current [23–28]. Mussels can adhere to the surface of the object by mucin secreted by the byssus [29–31]. The excellent adhesion properties of animals in nature have aroused great interest of researchers. Through a large number of in-depth studies on adhesive animals, the adhesion mechanism is revealed. At present, biological adhesion is classified into three main types: interlocking, friction and bonding. Interlocking is the use of an animal's toes or claws to hold tightly to the surface of an object at protruding, uneven areas or penetrate into the object's interior. Friction can be classified into microscale interlocking and friction force. Bonding can be further classified into four types: dry adhesion, wet adhesion, vacuum adhesion and chemical adhesion. The action mode of dry adhesion is the van der Waals force between the sucker and the object. Wet adhesion is the capillary force between the sucker and the object. Vacuum adhesion is the pressure difference between inside and outside the sucker. Chemical adhesion is a kind of protein colloid secreted by animals, which can effectively adhere to the surface of the object [32–34].

At present, researchers have carried out research on some animals with excellent adhesion, and achieved a lot of results, such as octopus and gecko. But the aquatic abalone with strong adhesion capacity has not been studied enough. Abalone is a kind of marine shellfish, usually living in offshore reef areas with low water temperature, and it adheres and moves by the abdominal foot [35, 36]. According to the research, an abalone with a shell length of about 15cm has an adhesion capacity of up to 200kg, which shows the strong adhesion force [37]. Due to the strong adhesion capacity of abalone abdominal foot, researchers explored the composition of abalone's adhesion capacity based on several types of animal adhesion characteristics [38]. Lin, et al. [39] measured the adhesion force of a single fiber on the surface of abalone abdominal foot by atomic force microscope, and calculated the van der Waals force of a single fiber theoretically, which was found to be consistent with the experimental results. The important role of van der Waals force and capillary force in the adhesion of abalone abdominal foot was also explained. Jing Li, et al. [40] measured the adhesion force and shear force between the abalone and different force plates in water and air, and showed that the adhesion force of abalone abdominal foot is mainly composed of vacuum adhesion force, van der Waals force and capillary force. The composition of abalone abdominal foot adhesion force is similar to several other typical adhesive creatures. Tim Kampowski, et al. found that the vacuum adhesion is the main part of the adhesion force of leech sucker, while capillary force, van der Waals force and chemical adhesion have enhanced the adhesion force [41]. Dylan K. Wainwright, et al. found that the adhesion force was mainly from the vacuum adhesion of the abdomen through the study of the sucker of the clingfish (*Gobiesocidae*) [42]. The micro-multistage structure of the sucker surface can make it better adapt to the surface of different roughness, form an effective internal locking structure and increase the friction force, reduce the probability of the sucker edge sliding inward when the clingfish is pulled, and improve the sealing performance. At present, researches on biological adhesion is mainly analyzed from the types of adhesion force and the corresponding mode of action. The adhesion of abalone was also studied by measuring the adhesion force of abdominal foot on different force measuring plates, and the composition and action mode of adhesion force were analyzed according to the test results. However, the

size of each type adhesive force in the total adhesive force of the abdominal foot and the primary and secondary roles played in the adhesion process have not been specifically analyzed and studied. Therefore, we studied the composition, size and corresponding mechanism of the adhesive force by observing abalone abdominal foot and measuring adhesion.

The main contents of this paper are as follows: firstly, the macroscopic and microscopic morphology of abalone abdominal foot surface was observed, and then the force measuring plates used to measure the adhesion force of abalone abdominal foot were designed and processed. The tensile test on the adhesion force of abalone on different force measuring plates was carried out. According to the test results, the types of adhesive force and corresponding sizes of abalone abdominal foot on different force measuring plates were calculated and analyzed, and the mechanism of different adhesive forces in abalone abdominal foot adhesion was discussed and elucidated. This study quantified the proportion of different adhesion forces in the adhesion of abalone, and clarified the primary and secondary roles of various adhesion forces when abalone adhered on the force plate with different surface morphologies. It provides a reference for the further study of other kinds of adhesive creatures and the design of bionic underwater adhesion devices.

## Materials and methods

### Preparation of abalone sample before test

The abalone used in the experiment was *Haliotis discus hannai*, which was purchased from Changchun, China. And quickly transferred to the laboratory aquarium. The size of the aquarium is 1500 mm × 600 mm × 600 mm, with a filtration system and a water circulation system. The water temperature in the aquarium is controlled between 15˚C and 20˚C, the salinity is 30%, the water depth was 0.8 m, and the bait is undaria, so as to ensure its normal survival in the aquarium. The mass of abalone selected in the experiment is generally between 50 g and 65 g, and the corresponding abdominal foot area was 1915 mm$^2$ to 2760 mm$^2$. Before the experiment, abalone was raised in the aquarium for at least 10 days, in order to reduce the difference of abalone individuals caused by different living environment and reduce the test error.

### The image of abalone abdominal foot

**Macroscopic observation.** Abdominal foot is the main executive organ of abalone adhesion and crawling. The surface morphology of abdominal foot was observed by stereomicroscope (ZEISS Stemi 2000-C) to prepare for further study on abalone adhesion. The surface morphology of the abalone abdominal foot is shown in Fig 1(A), where the surface of abdominal foot can be divided into three layers, namely the outer layer, the middle layer and the inner layer. The inner layer occupies most of the abdominal foot and has a large number of striped folds on the surface, as shown in Fig 1(B). The abalone moves forward through the driving force generated by expansion and contraction of the folds in different regions of the abdominal foot.

**Microscopic observation.** In order to observe the micro-morphology of abalone abdominal foot surface, scanning electron microscope (SEM) was used to observe it. First, the abalone in the adhesion state was anesthetized to keep the abdominal foot in the unfolded state, so as to extract small sample of the abdominal foot for observation. The surface tissue of abalone abdominal foot (5 mm × 5 mm × 3 mm) was cut and fixed in 2.5% glutaraldehyde solution for about 2 hours. Then the fixed sample was rinsed with phosphate buffer (PBS) for 3 times, each time for about 15min. The sample was fixed in osmic acid, and the temperature was set at 4˚C for about 2 hours. The sample was rinsed again with phosphate buffer (PBS) for 3 times, each time for about 10 min. The sample was dehydrated gradually with 50%, 70%, 80%, 90%, 100%

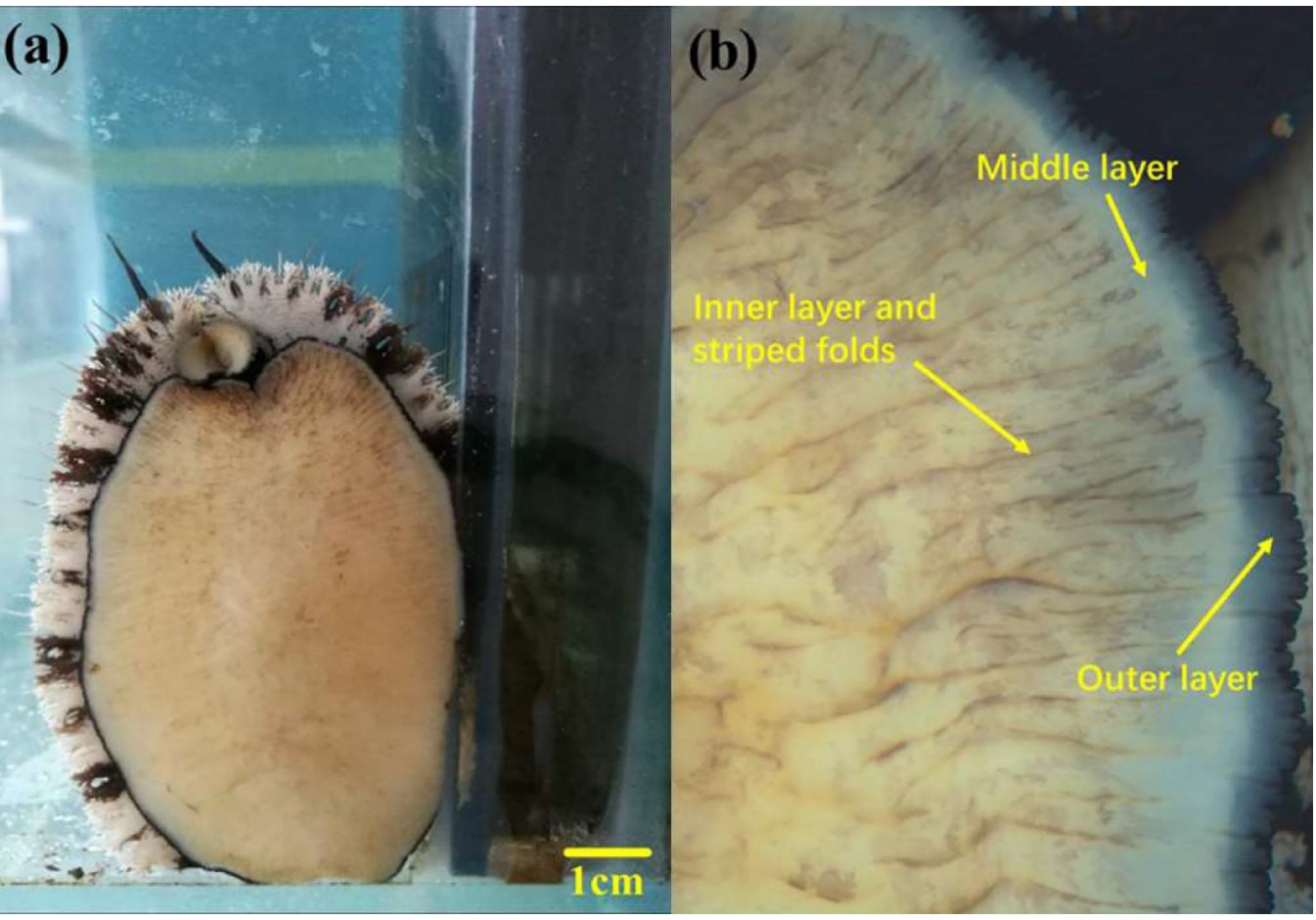

**Fig 1.** (a) Surface morphology of abalone abdominal foot (b) Three layers and striped folds of abalone abdominal foot.

ethanol, in which 50%, 70%, 80% and 90% ethanol were dehydrated once, each time for 10 min, 100% ethanol was dehydrated 3 times, each time for 10 min. Finally, the sample was put into a vacuum freeze dryer for freeze-drying. The surface of dried abdominal foot sample was sprayed with gold, and then observed by scanning electron microscope (JSM-5500LV).

## Preparation of force measuring plates for abalone adhesion test

In order to study the composition of abalone abdominal foot adhesion force and the proportion of various adhesion force to the total adhesion force, it is necessary to conduct an adhesion tensile test on abalone. According to the observation and related research on abalone adhesion, the adhesion force of abalone abdominal foot is mainly composed of vacuum adhesion force, van der Waals force and capillary force. Based on the composition of the abdominal foot adhesion force, five types of force measuring plates were designed and manufactured for the adhesion test of abalone abdominal foot. The five types of force measuring plates are: 1. Smooth acrylic plate 2. Through-hole acrylic plate 3. Striped grooved acrylic plate 4. Smooth Teflon plate 5. Through-hole Teflon plate. The five force measuring plates have the same size, with a diameter of 160 mm and a thickness of 10 mm. The size of the through-hole acrylic plate is the same as that of the through-hole Teflon plate, in which the hole diameter is 3 mm, the hole spacing is 8 mm, and the angle between the connecting hole line is 60˚. The specific

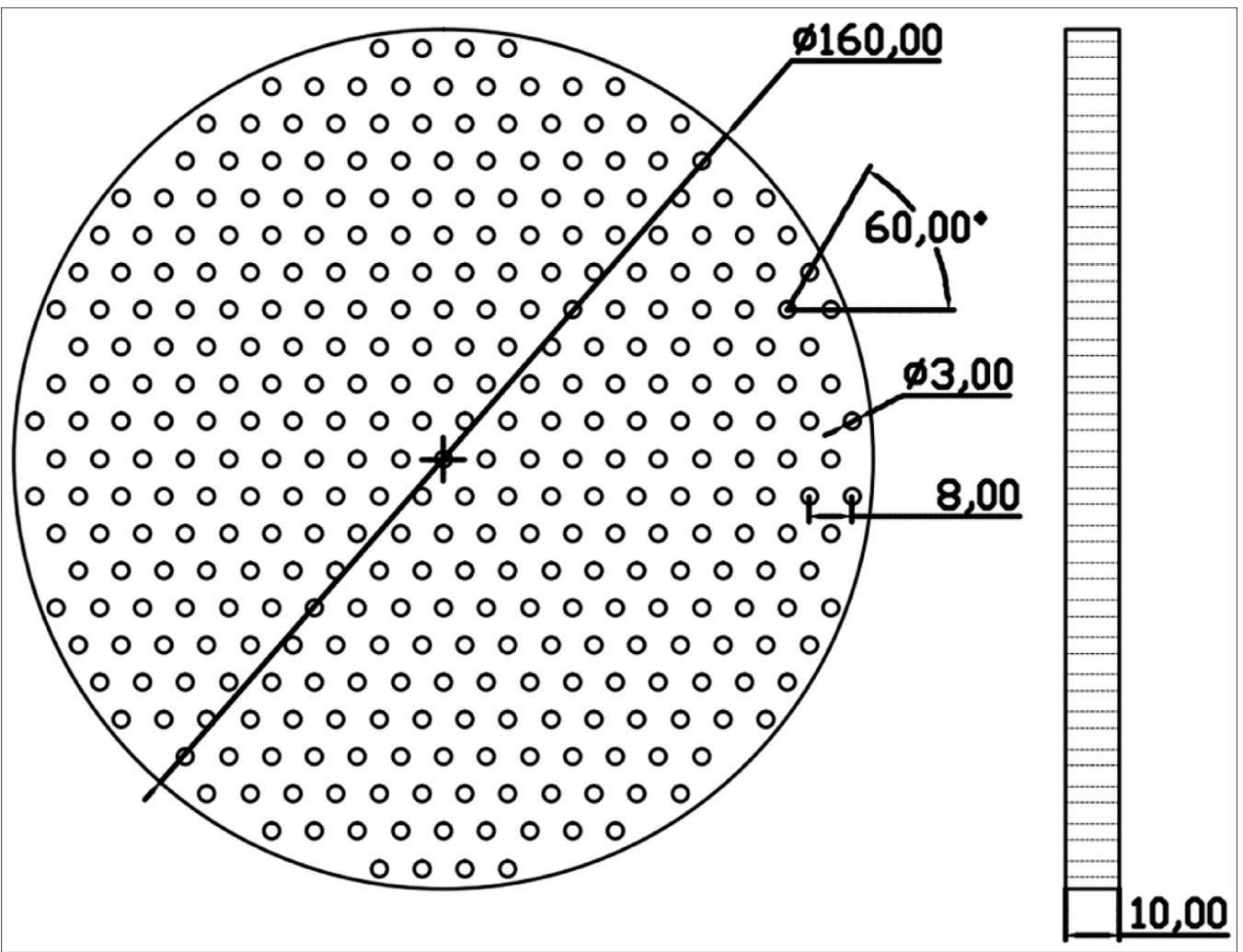

**Fig 2. The diagram of through-hole measuring plate.**

design dimensions are shown in Fig 2. The design dimensions of the striped grooved acrylic plate are shown in Fig 3, in which the groove width is 2 mm, the spacing is 8 mm, and the depth is 5 mm. The five types of force measuring plates after manufactured are shown in Fig 4.

## Tensile test of abalone

The universal testing machine used in tensile test is WSM-500N, which is controlled by computer. Before the test, five types of force measuring plates were placed at the bottom of the leaching tub, and then five abalones were placed on five types of force measuring plates for slow adhesion, are shown in Fig 5. The role of the leaching tub is to ensure the flowing water environment needed for abalone survival, while preventing abalone from moving outside the force measuring plate. In each test, the force measuring plate and abalone adhere on its surface are put on the test machine. During the test, the force measuring plate is fixed first, and then the abalone shell is hooked with the hook. The lifting speed of the tensile testing machine is 100 mm/min. And the test period was 24 hours to ensure that the adhesion force of abalone did not increase with time.

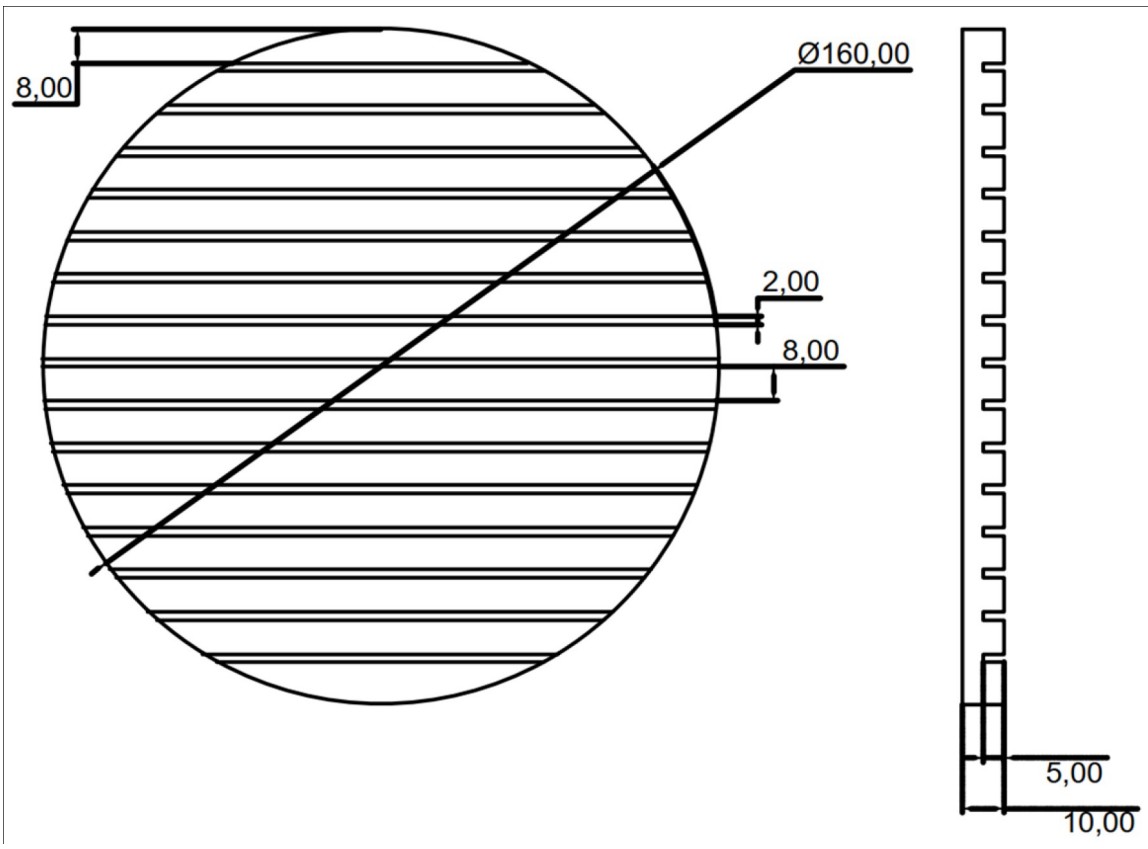

**Fig 3. The diagram of striped grooved measuring plate.**

## Results

### Microscopic morphology of abalone abdominal foot

The surface microscopic morphology of abalone abdominal foot is shown in Fig 6(A)–6(C). From Fig 6(A) and 6(B), it can be seen that the surface of abdominal foot is covered with a large number of fibers, the direction is perpendicular to the surface of the abdominal foot, and the fibers are independent of each other with gaps in the middle. Fig 6(C) shows a magnified view of the fibers, it can be seen from the figure that the height of the fiber is basically the same, ranging from 35 to 45 μm. The root of the fiber is thinner, the top is thicker, and the diameter is 0.5 to 4 μm.

### Tensile test results of abalone

The adhesion force and corresponding mass of abalone on the five force measuring plates are shown in Table 1. In order to analyze the adhesion force of abalone abdominal foot on different force measuring plates, it is necessary to calculate the corresponding adhesion strength $f$, that is, the adhesion force $F$ of abalone on the force measuring plate is divided by the corresponding abdominal foot area $A$, in which $f = F/A$. Because it is difficult to measure the adhesion area of abdominal foot in the test, in order to obtain the adhesion area of abdominal foot, the method of measuring and calculating the mass of abalone is used. Before the test, 10 abalones were selected and their mass(g) and corresponding abdominal foot area(mm$^2$) were measured respectively, and then the ratio R of abdominal foot area(mm$^2$) to mass(g) of each

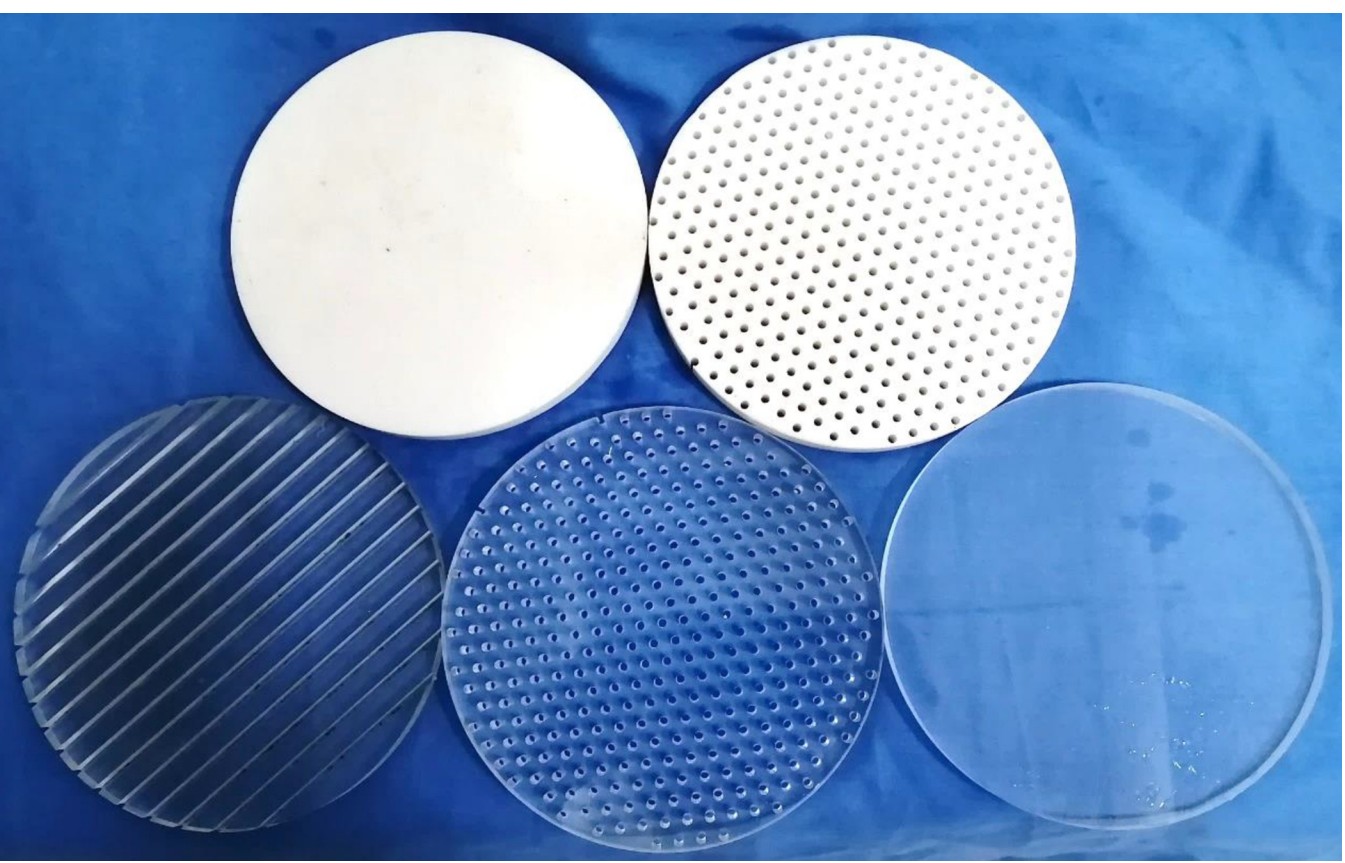

**Fig 4. The five types of force measuring plates after manufactured.**

abalone was calculated, the average value of ratio R of 10 abalones was 43.15. In this way, the corresponding abdominal foot area can be calculated by measuring the mass of abalone. The adhesion strength *f* of abalone on five types of force measuring plates is calculated from Table 1, the results are shown in Table 2. The column chart of the average adhesion strength *f* is shown in Fig 7. It can be seen from the Fig 7 that the average adhesion strength *f* of abalone on the smooth acrylic plate is the largest, followed by the smooth Teflon plate. The adhesion strength *f* of the through hole or striped grooved acrylic plate is smaller than that of the smooth acrylic plate, which is consistent with the adhesion characteristics of abalone. The through-hole Teflon plate has the smallest average adhesion strength.

## Calculation of various adhesive forces in abalone tensile test

**Calculation of van der Waals force.** In order to calculate the proportion of various adhesion forces to the total adhesion force of abalone abdominal foot, it is necessary to calculate the vacuum adhesion force, van der Waals force and capillary force generated by abalone abdominal foot. Because the vacuum degree between the abdominal foot and the force measuring plate is unknown during abalone adhesion, it is assumed that the vacuum degree of the abdominal foot is 100%. Through the calculation, the adhesion force of abdominal foot caused by the vacuum action is much larger than the test results in Table 1, indicating that the vacuum degree of the abdominal foot is far less than 100%. Therefore, the van der Waals force generated by the abdominal foot is calculated first. From the microscopic observation of the abalone

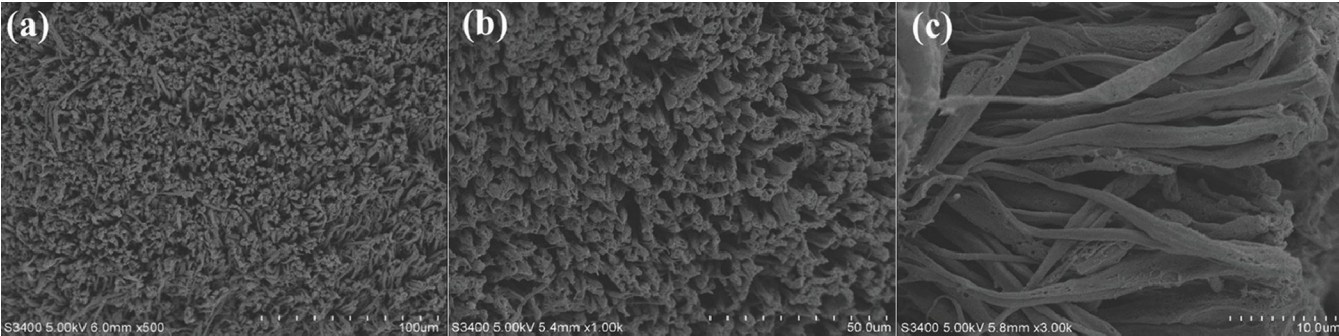

**Fig 5. Abalone adhered on five types of force measuring plates before test.**

abdominal foot surface in Fig 6, it can be seen that the surface is composed of a large number of micron-sized fibers. Each fiber can form a van der Waals force when it comes into contact with the force measuring plate. In this paper, the van der Waals force of a single fiber is calculated at first, and then the van der Waals force produced by the whole abdominal foot surface is calculated according to the number of fibers per unit area. According to the JKR form of the

**Fig 6.** (a) SEM image of a mass of fibers on the abdominal foot (b) Localized SEM image of the abdominal foot at high magnification (c) SEM image of several fibers on the abdominal foot.

Table 1. Abalone adhesion force (*F*) on five types of force measuring plates and corresponding mass.

| Maximum adhesion force (N) | Force measuring plate type | | | | |
|---|---|---|---|---|---|
| Number of test | Smooth acrylic | Through-hole acrylic | Striped grooved acrylic | Smooth Teflon | Through-hole Teflon |
| 1 | 109.7 | 44.51 | 54.6 | 96.33 | 38.34 |
| Abalone mass (g) | 60.8 | 39.4 | 50.5 | 53.6 | 55.1 |
| 2 | 112 | 64.53 | 34.61 | 80.78 | 35 |
| Abalone mass (g) | 58.8 | 60 | 35.8 | 50.4 | 58.6 |
| 3 | 122.5 | 71.3 | 62 | 48.25 | 75.68 |
| Abalone mass (g) | 56.5 | 51.7 | 38.1 | 54.9 | 54.8 |
| 4 | 93.1 | 48.95 | 47.47 | 61.62 | 50.31 |
| Abalone mass (g) | 55.5 | 56.7 | 37.5 | 54.9 | 59.2 |
| 5 | 108.1 | 41 | 104.2 | 75.48 | 50.04 |
| Abalone mass (g) | 60.5 | 35.3 | 54 | 53.4 | 56.6 |
| 6 | 119.8 | 59.32 | 54.28 | 78.75 | 55.07 |
| Abalone mass (g) | 63.4 | 57.2 | 63 | 53.9 | 50.6 |
| 7 | 88.46 | 63.34 | 104.9 | 111.8 | 40.66 |
| Abalone mass (g) | 55.5 | 51.4 | 51.4 | 54.9 | 48 |
| 8 | 100 | 67.74 | 80.82 | 69.61 | 53.74 |
| Abalone mass (g) | 58 | 53.2 | 55.1 | 53.1 | 58 |
| 9 | 110.5 | 73.76 | 45 | 77.51 | 37.05 |
| Abalone mass (g) | 53.3 | 54 | 45.8 | 52.1 | 52.9 |
| 10 | 106 | 80.61 | 54.2 | 76.08 | 49.32 |
| Abalone mass (g) | 51.9 | 54.5 | 73.8 | 50.2 | 54.5 |

pull-off force equation, the formula of single fiber adhesion force *P* is derived as follows:

$$p = \frac{3}{2}\pi R\Delta\gamma \tag{1}$$

And the surface energy $\Delta\gamma$

$$\Delta\gamma = \frac{A}{24\pi D_0^2} \tag{2}$$

$$P = \frac{AR}{16D_0^2} \tag{3}$$

Table 2. Abalone adhesion strength (*f*) on five types of force measuring plates.

| Adhesion strength(kPa) | Force measuring plate type | | | | |
|---|---|---|---|---|---|
| Number of test | Smooth acrylic | Through-hole acrylic | Striped grooved acrylic | Smooth Teflon | Through-hole Teflon |
| 1 | 41.81 | 26.18 | 25.06 | 41.65 | 16.13 |
| 2 | 44.14 | 24.92 | 22.40 | 37.14 | 13.84 |
| 3 | 50.25 | 31.96 | 37.71 | 20.37 | 32.01 |
| 4 | 38.88 | 20.01 | 29.34 | 26.01 | 19.69 |
| 5 | 41.41 | 26.92 | 44.72 | 32.76 | 20.49 |
| 6 | 43.79 | 24.03 | 19.97 | 33.86 | 25.22 |
| 7 | 36.94 | 28.56 | 47.30 | 47.19 | 19.63 |
| 8 | 39.96 | 29.51 | 33.99 | 30.38 | 21.47 |
| 9 | 48.05 | 31.66 | 22.77 | 34.48 | 16.23 |
| 10 | 47.33 | 34.28 | 17.02 | 35.12 | 20.97 |
| Average value | 43.26 | 27.8 | 30.03 | 33.9 | 20.57 |

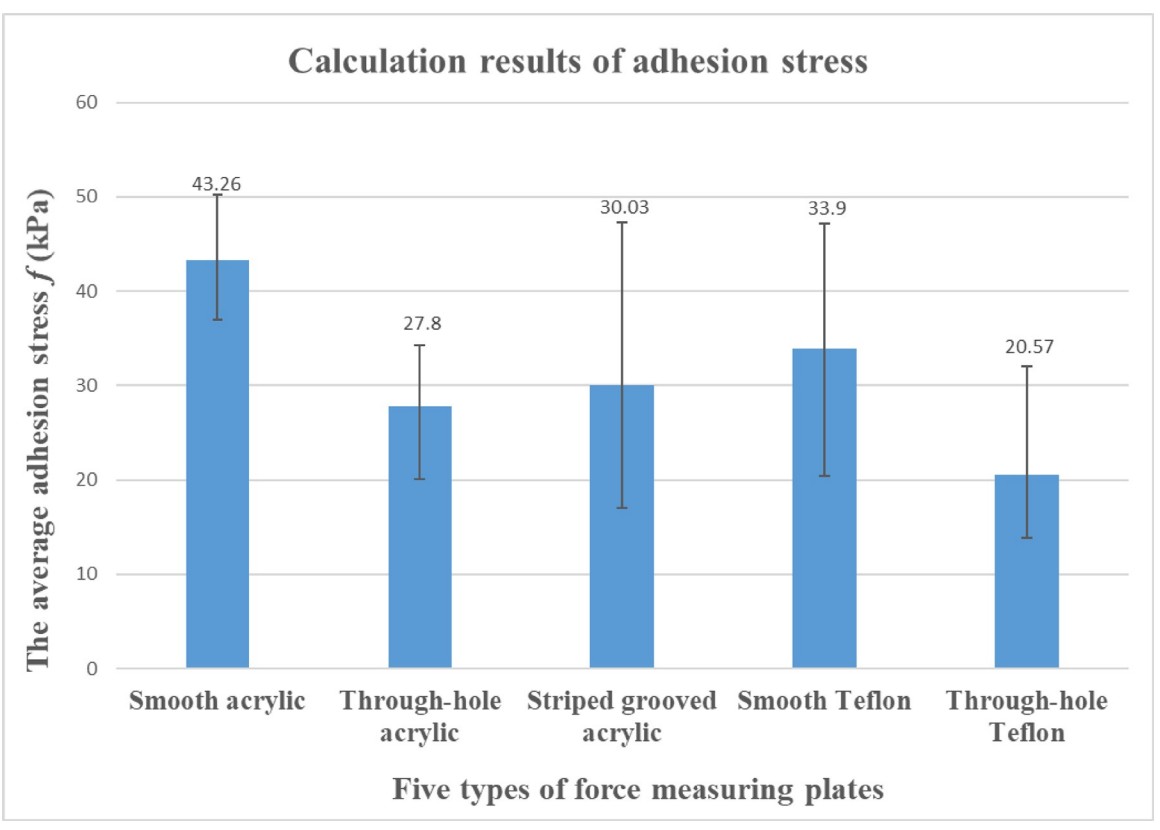

**Fig 7. Average adhesion strength *f* of abalone on five force measuring plates.**

*A* is the Hamaker constant in the formula, and its value is about $6 \times 10^{-2}$ J, $D_0 = 0.165$ m. It is a constant, which is suitable for most materials. *R* is the abdominal foot fiber radius. It can be seen from Fig 6 that the fiber radius *R* is about 1 to 1.5 μm. Therefore, the van der Waals force formed by a single fiber and the force measuring plate is about 135~200 nN. The area of $50 \times 50$ μm is selected in Fig 6, and the number of fibers to be about 120–140. When calculating the van der Waals force of the abalone on the through hole and striped groove force measuring plate, the corresponding non-contact part needs to be removed. As a result, the average van der Waals force of abalone abdominal foot on different force measuring plates is calculated, as shown in Table 3.

**Measurement and calculation of capillary force.**   In the tensile test, when the abalone and the force measuring plate are taken out of the aquarium, the presence of water between the abdominal foot and the force measuring plate will produce capillary force. In order to measure the capillary force between the abdominal foot and force measuring plate, abalone that has just died (the dead abalone abdominal foot does not produce vacuum adhesion force and van der Waals force) with fully expanded abdominal foot was selected to measure the capillary force *F* on the smooth acrylic and the smooth Teflon plate. Then the corresponding adhesion strength *f* was calculated. The specific test results are shown in Tables 4 and 5.

The capillary force of abalone abdominal foot can also be obtained by theoretical calculation. Because abalone adhesion belongs to the capillary type between two parallel plates. According to capillary theory model, the capillary force is composed of two parts, one is the force of the negative pressure in the liquid on the axis of the solid wetted area, the other is the force of the surface tension on the liquid side surface on the solid, liquid and gas three-phase

**Table 3.  The average van der Waals force on abalone abdominal foot (unit: N).**

| Number of test | The average van der Waals force (N) | | | | |
|---|---|---|---|---|---|
| | Smooth acrylic | Through-hole acrylic | Striped grooved acrylic | Smooth Teflon | Through-hole Teflon |
| 1 | 23.19 | 12.47 | 15.41 | 20.45 | 18.46 |
| 2 | 22.43 | 20.01 | 10.96 | 19.22 | 19.48 |
| 3 | 21.55 | 17.16 | 11.69 | 20.94 | 18.34 |
| 4 | 21.17 | 18.76 | 11.48 | 20.94 | 19.71 |
| 5 | 23.08 | 10.78 | 16.49 | 20.37 | 18.90 |
| 6 | 24.18 | 19.45 | 19.20 | 20.56 | 16.93 |
| 7 | 21.17 | 17.42 | 15.65 | 20.94 | 16.12 |
| 8 | 22.12 | 17.48 | 16.80 | 20.25 | 19.31 |
| 9 | 20.33 | 18.04 | 13.99 | 19.87 | 17.62 |
| 10 | 19.80 | 18.23 | 22.53 | 19.15 | 18.23 |

line along the axis of the liquid bridge. According to the Young-Laplace equation, the calculation formula of capillary force:

$$F = \gamma A_a \left( \frac{\cos\theta_1 + \cos\theta_2}{d} + \frac{1}{r_2} \right) + \gamma l_a \sin\theta_1 \tag{4}$$

$\gamma$ is the surface tension of liquid, which is 73 mN/m, $A_\alpha$ is the area of the solid wetted by the liquid, $\theta_1$ and $\theta_2$ are the contact angles between the abdominal foot and the force measuring plate. The contact angles of abalone foot, acrylic plate and Teflon plate were 0°,85° and 114° respectively, $r_2$ is taken as 28, $l_\alpha$ is the circumference of the wetted area of the abalone, taken as 175 mm. Because the distance $d$ between the abdominal foot and the force measuring plate is unknown, the capillary force $F$ cannot be calculated. According to the test results in Table 4 and combined with Formula 2, the distance $d$ between the abdominal foot and the force measuring plate was calculated to be 169 μm. Thus, it can be calculated that the capillary force of abalone on smooth Teflon plate is 0.6 N, and the corresponding adhesion strength is 0.26 kPa. This is basically the same with the measured capillary force value of 0.27 kPa of abalone on the smooth Teflon plate in Table 5, indicating that the theoretical calculation is consistent with the actual measurement results.

**Calculation of vacuum adhesion force.**   Since the adhesion force of abalone abdominal foot is mainly composed of vacuum adhesion force, van der Waals force and capillary force. The capillary force is no more than 1 N according to the adhesion test and calculation. And the proportion of capillary force is very small, which can be ignored. Therefore, the vacuum adhesion force of the abdominal foot on different force measuring plates can be obtained by subtracting the van der Waals force from the total adhesion force, and the vacuum adhesion strength and the vacuum degree of abalone on five types of force measuring plates can be calculated, as shown in Table 6.

**Table 4.  Capillary force of abalone abdominal foot on smooth acrylic plate.**

| Number of test | Abalone mass(g) | Capillary force (N) | Capillary strength (kPa) |
|---|---|---|---|
| 1 | 44.6 | 0.807 | 0.42 |
| 2 | 44.6 | 1.163 | 0.60 |
| 3 | 44.6 | 0.799 | 0.42 |
| 4 | 51.3 | 0.975 | 0.44 |
| Average value | 46.275 | 0.936 | 0.47 |

**Table 5. Capillary force of abalone abdominal foot on smooth Teflon plate.**

| Number of test | Abalone mass(g) | Capillary force (N) | Capillary strength (kPa) |
|:---:|:---:|:---:|:---:|
| 1 | 58.3 | 0.636 | 0.25 |
| 2 | 52.2 | 0.697 | 0.31 |
| 3 | 52.2 | 0.55 | 0.25 |
| 4 | 49.5 | 0.673 | 0.32 |
| 5 | 52.6 | 0.483 | 0.21 |
| Average value | 52.96 | 0.61 | 0.27 |

**The proportion of various adhesive forces in total adhesive force of abalone.** In order to calculate the proportion of various adhesive forces in abdominal foot total adhesive force of abalone, the van der Waals force of abalone on the five force plates in Table 3 is divided by the total adhesive force of abalone on the five force measuring plates in the corresponding position in Table 1, and the results are averaged to obtain the proportion of van der Waals force in the total adhesive force. After removing the proportion of van der Waals force in the total adhesive force, the proportion of other adhesive forces can be obtained. The capillary force stress of abalone on smooth acrylic plate is 0.47 kPa, while that on smooth Teflon plate is 0.27 kPa. The proportion of capillary force can be calculated, which accounts for about 1% of the total adhesive force.

According to the results in Table 6, abalone has vacuum adhesion on smooth acrylic, through-hole acrylic and striped groove acrylic force measuring plate, but the vacuum degree on through-hole acrylic and striped groove acrylic plate is smaller than that of smooth acrylic plate, the vacuum degree has certain loss. The loss of the vacuum adhesion force of abdominal foot on the through-hole acrylic plate is due to the existence of the through-hole. Because the abdominal foot has a certain degree of elasticity, the through-hole can be blocked, so that the abalone abdominal foot can form a whole sucker. Therefore, it can be inferred that the loss of vacuum adhesion force is due to the destruction of vacuum adhesion in the local area of the abdominal foot by the through-hole. The loss of vacuum adhesion force of abdominal foot on the striped groove acrylic plate is due to the existence of the striped groove. Because the striped groove destroys the adhesion of the abalone abdominal foot as a whole sucker, but the vacuum adhesion generated by the local area of the abdominal foot is still present. The vacuum

**Table 6. The vacuum adhesion strength of abalone abdominal foot.**

| Adhesion strength (kPa) | Force measuring plate type | | | | |
|:---:|:---:|:---:|:---:|:---:|:---:|
| Number of test | Smooth acrylic | Through-hole acrylic | Striped grooved acrylic | Smooth Teflon | Through-hole Teflon |
| 1 | 32.97 | 18.85 | 17.98 | 32.81 | 8.36 |
| 2 | 35.30 | 17.20 | 15.31 | 28.31 | 6.14 |
| 3 | 41.41 | 24.27 | 30.60 | 11.53 | 24.25 |
| 4 | 30.04 | 12.34 | 22.24 | 17.17 | 11.98 |
| 5 | 32.57 | 19.84 | 37.64 | 23.92 | 12.75 |
| 6 | 34.95 | 16.15 | 12.90 | 25.02 | 17.47 |
| 7 | 28.10 | 20.70 | 40.24 | 38.35 | 11.85 |
| 8 | 31.12 | 21.89 | 26.93 | 21.54 | 13.76 |
| 9 | 39.21 | 23.91 | 15.69 | 25.64 | 8.51 |
| 10 | 38.49 | 26.53 | 9.95 | 26.28 | 13.22 |
| Average value | 34.42 | 20.17 | 22.95 | 25.06 | 12.83 |
| Vacuum degree | 33.97% | 19.90% | 22.65% | 24.73% | 12.66% |

Table 7. The percentage of various adhesive forces of abalone abdominal foot on five types of force measuring plates.

| The percentage of adhesion strength (%) | | Force measuring plate type | | | | |
|---|---|---|---|---|---|---|
| Types of adhesive forces | | Smooth Acrylic | Through-hole Acrylic | Striped grooved Acrylic | Smooth Teflon | Through-hole Teflon |
| Vacuum adhesion force | The whole abdominal foot | 28.17 | 43.85 | 0 | 35.9 | 57.19 |
| | The local abdominal foot | 27.00 | 26.37 | 72.18 | 34.42 | 1.92 |
| | Friction force | 23.12 | | | 1.47 | |
| Capillary force | | 1.09 | 1.69 | 1.57 | 0.8 | 1.31 |
| Van der Waals force | | 20.62 | 28.09 | 26.25 | 27.41 | 39.58 |

adhesion of smooth acrylic plate is obviously larger than that of smooth Teflon plate, mainly because the friction coefficient of Teflon plate is extremely small. When abalone is pulled upward, friction can slow down the possibility of leakage due to the inward contraction of the edge of the abdominal foot, which indirectly improves the vacuum adhesion of the abdominal foot.

It can be seen that the vacuum adhesion of abalone abdominal foot can be further divided into the whole adhesion of abdominal foot, the local adhesion of abdominal foot and the frictional equivalent vacuum adhesion. Because the friction coefficient of smooth Teflon plate is very small, only 0.04, which is only one-twentieth of smooth acrylic plate. Therefore, by calculating the difference between the vacuum adhesion of abalone on smooth acrylic plate and that on smooth Teflon plate, the frictional equivalent vacuum adhesion of abalone on smooth acrylic plate can be obtained. Since the abalone abdominal foot does not have the whole vacuum adhesion on the striped grooved acrylic plate, the local vacuum adhesion of abdominal foot can be obtained by subtracting the frictional equivalent vacuum adhesion from the vacuum adhesion of abalone on the striped grooved acrylic plate. Thus, the whole adhesion of abdominal foot on the smooth acrylic plate can be calculated. Therefore, the proportion of various adhesion forces on the five force measuring plates to the total adhesion forces of abdominal foot can be analyzed and calculated, as shown in Table 7, and the corresponding column chart is shown in Fig 8.

## Discussion

According to Table 7 and Fig 8, the vacuum adhesion force of the five force measuring plates accounts for more than half of the total adhesion force of abalone abdominal foot, and van der Waals force also plays an important role in the adhesion of abalone abdominal foot. However, the proportion of capillary force in abalone abdominal foot adhesion is very small, about 1%. The vacuum adhesion of abdominal foot can be divided into three parts: namely the whole of abdominal foot, the local part of abdominal foot and the frictional equivalent vacuum adhesion, as shown in Fig 9.

In the vacuum adhesion of abalone, the vacuum adhesion force generated by the whole sucker which formed between the whole abdominal foot and the force measuring plate is basically equivalent to the vacuum adhesion force generated by the local part of abdominal foot. When abalone adheres on the irregular surface, the elasticity of abalone abdominal foot is difficult to seal with the adhesion surface to form a whole sucker structure, and the abalone can adhere by van der Waals force and the vacuum adhesion force generated by the local part of abdominal foot, as shown in Fig 10(A). When abalone adhered on the surface with holes, the elasticity of the abdominal foot can block the holes, thus forming a whole sucker structure, and the abalone adheres by van der Waals force and the vacuum action formed by the abdominal foot whole sucker, as shown in Fig 10(B).

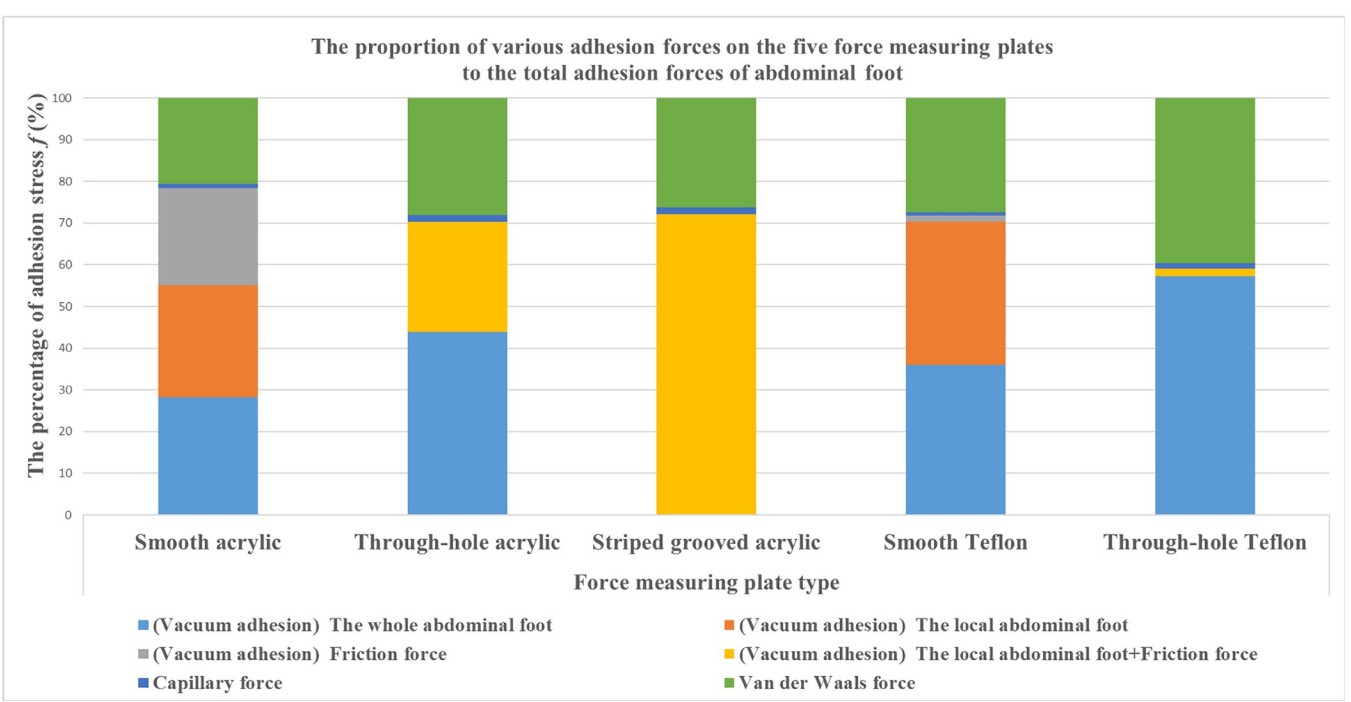

**Fig 8. The proportion of various forces in the adhesion force of abalone abdominal foot.**

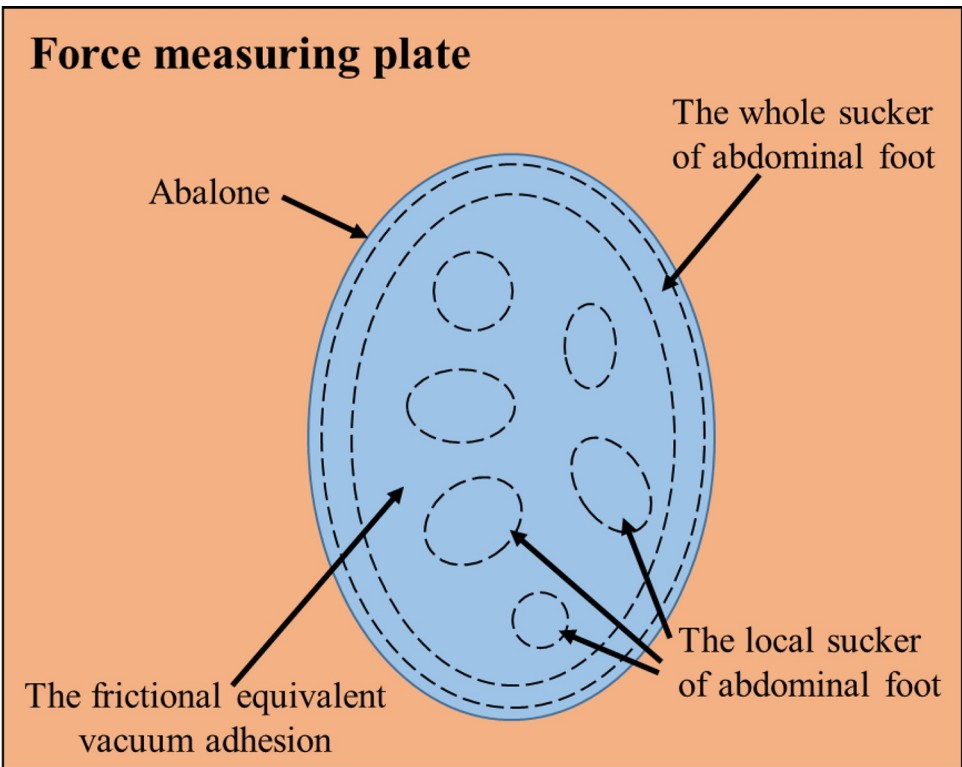

**Fig 9. The whole abdominal foot vacuum adhesion, the local part of abdominal foot vacuum adhesion and the frictional equivalent vacuum adhesion on the force measuring plate.**

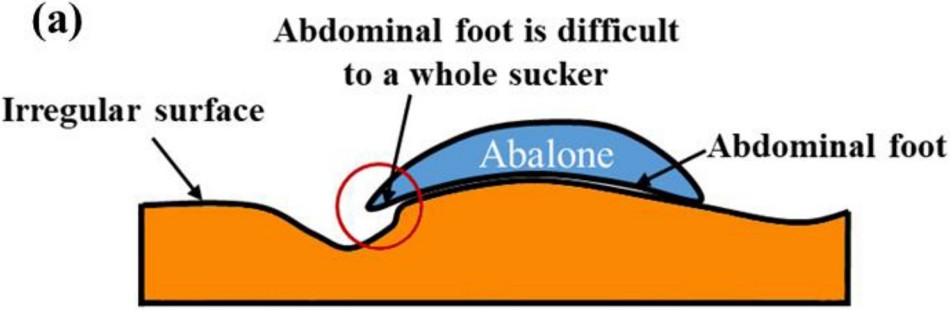

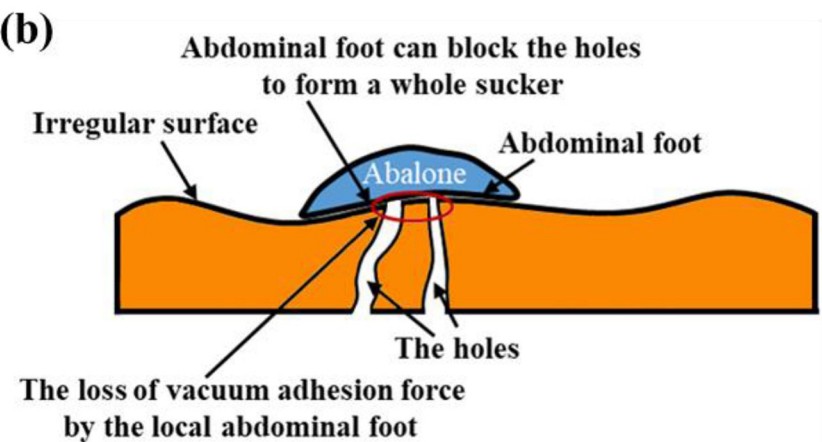

**Fig 10.** (a) The abalone abdominal foot is difficult to seal with the adhesion surface to form a whole sucker structure. (b) The abalone adhered on the surface with holes.

When the friction coefficient of the adhesion surface is not too small, the friction between the abdominal foot and the adhesion surface can effectively slow down the possibility of leakage caused by the inward contraction of the abdominal foot edge, which has an equivalent vacuum adhesion effect, and its size is equivalent to van der Waals force, as shown in Fig 11(A). The proportion of capillary force in the total adhesion force of abalone is very small, and it has little effect on the actual adhesion of abdominal foot. However, the liquid film formed by the capillarity can fill the gap between the abdominal foot and the adhesion surface, prevent the gas from flowing into the sucker, and indirectly improve the adhesion capacity of the abdominal foot, as shown in Fig 11(B).

The characteristics of various adhesion forces in abalone abdominal foot can effectively improve its adaptability on different adhesion surfaces, so that it can better adapt to the natural environment, which is very important for its survival. This study quantifies the proportion of various adhesion forces to the total adhesion force of the abdominal foot, and the roles of each adhesion force under different adhesion conditions were analyzed. Although there are some issues need to be further discussed, such as the role of the small amount of abdominal foot mucus in adhesion, the calculation of van der Waals force is relatively ideal conditions, and the adhesion force of abalone changes in different environments. We hope our analysis provides a reference for the further study of other adhesive creatures and the design of bionic underwater adhesion devices.

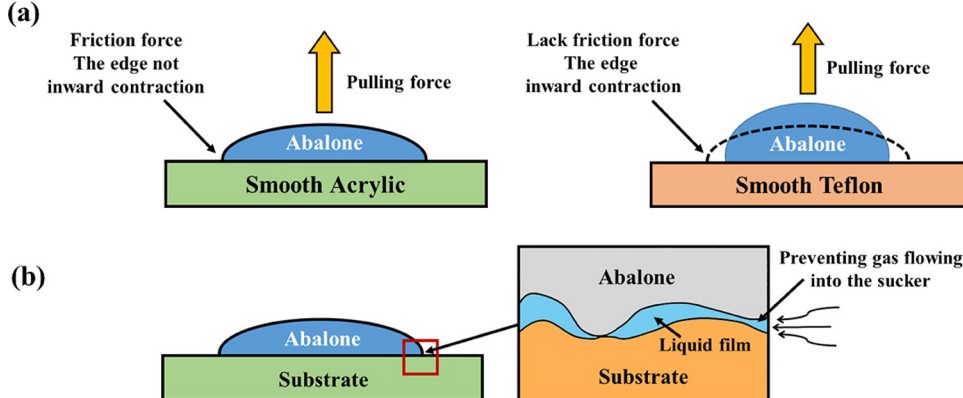

**Fig 11.** (a) The frictional equivalent vacuum adhesion effect. (b) The liquid film formed by the capillarity indirectly improve the adhesion capacity of the abdominal foot.

## Conclusions

In this study, the macroscopic and microscopic morphology of the abalone abdominal foot surface was observed, then five types of force measuring plates were designed and processed for the adhesion test of abalone abdominal foot. According to the test results, the proportion of various adhesion forces to the total adhesion force in abalone abdominal foot was calculated and analyzed. The main conclusions are as follows:

1. The surface of abalone abdominal foot consists of a large number of equal-height and densely arranged cylindrical fibers, which are perpendicular to the surface of the abdominal foot. The diameter of the fibers is 0.5–4 μm, the height is 35–45 μm.

2. The adhesion force of abalone abdominal foot is composed of vacuum adhesion force, van der Waals force and capillary force. The vacuum adhesion force accounts for more than half of the total adhesion force of abalone abdominal foot, and van der Waals force also plays an important role in the adhesion of abalone abdominal foot. The proportion of capillary force is very small, and its main role is to form a liquid film to prevent the gas from flowing into the sucker, and indirectly improve the adhesion capacity of the abdominal foot.

3. The vacuum adhesion of abalone abdominal foot can be divided into three parts: namely the whole adhesion of abalone abdominal foot, the local adhesion of abdominal foot and the frictional equivalent vacuum adhesion. Among them, the vacuum adhesion of the whole abdominal foot is basically equivalent to that of the local abdominal foot.

## Acknowledgments

We thank Jin Xu for his assistance in tensile test.

## Author Contributions

**Conceptualization:** Peng Xi.

**Data curation:** Peng Xi.

**Formal analysis:** Peng Xi.

**Investigation:** Peng Xi, Shaobo Ye.

**Project administration:** Qian Cong.

**Resources:** Peng Xi.

**Supervision:** Peng Xi.

**Writing – original draft:** Peng Xi.

**Writing – review & editing:** Peng Xi.

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
