## [Editor Report · Decision Letter 0]

25 Aug 2022

PONE-D-22-19954Abalone adhesion: the role of various adhesion forces and proportion of total adhesion forcePLOS ONE

Dear Dr. xi,

Thank you for submitting your manuscript to PLOS ONE. After careful consideration, we feel that it has merit but does not fully meet PLOS ONE’s publication criteria as it currently stands. Therefore, we invite you to submit a revised version of the manuscript that addresses the points raised during the review process.

We look forward to receiving your revised manuscript.

Kind regards,

Jianguo Wang, PhD

Academic Editor

PLOS ONE

Journal Requirements:

Additional Editor Comments:

I read through the manuscript and found that this is an interesting investigation. But the manuscript should be improved on the following issues before it is sent out for review.

1, Please follow the Journal style of Plos One for writting-up, references and citations in the context.

2, Some subtitles are too general. Please use more exact subtitles to reflect the context in that section. All subtitles should be consistent in the whole manuscript

3, Please clearly define your novelty or contributions of this manuscript at the right positions. Please indicte the sources of formulae if necessary and state the calculation procedure.

4, Current reference list seems too old. The newest ones are in 2018. Please update the reference list by adding more references within the last three years. These references can help you to define the problem.
---

## [Author Response · Author response to Decision Letter 0]

4 Oct 2022

Response to Reviewers

Additional Editor Comments:

1. Please follow the Journal style of Plos One for writing-up, references and citations in the context.

I have revised writing-up, references and citations in the manuscript according to the Journal style of Plos One.

2. Some subtitles are too general. Please use more exact subtitles to reflect the context in that section. All subtitles should be consistent in the whole manuscript

I have revised some subtitles.

3. Please clearly define your novelty or contributions of this manuscript at the right positions. Please indicate the sources of formulae if necessary and state the calculation procedure.

I have defined my novelty or contributions of this manuscript at the right positions.

I have added the sources of formulae and state the calculation procedure at the right positions of manuscript.

4. Current reference list seems too old. The newest ones are in 2018. Please update the reference list by adding more references within the last three years. These references can help you to define the problem.

I have added more references within the last three years and update the reference list.

---

## [Decision Letter · Decision Letter 1]

31 Jan 2023

PONE-D-22-19954R1Abalone adhesion: the role of various adhesion forces and their proportion to total adhesion forcePLOS ONE

Dear Dr. xi,

Thank you for submitting your manuscript to PLOS ONE. After careful consideration, we feel that it has merit but does not fully meet PLOS ONE’s publication criteria as it currently stands. Therefore, we invite you to submit a revised version of the manuscript that addresses the points raised during the review process.

We look forward to receiving your revised manuscript.

Kind regards,

Arumugam Sundaramanickam, PhD

Academic Editor

PLOS ONE

Journal Requirements:

Reviewers' comments:

Reviewer's Responses to Questions

**Comments to the Author**

1. If the authors have adequately addressed your comments raised in a previous round of review and you feel that this manuscript is now acceptable for publication, you may indicate that here to bypass the “Comments to the Author” section, enter your conflict of interest statement in the “Confidential to Editor” section, and submit your "Accept" recommendation.

Reviewer #1: (No Response)

2. Is the manuscript technically sound, and do the data support the conclusions?

Reviewer #1: (No Response)

3. Has the statistical analysis been performed appropriately and rigorously? 

Reviewer #1: (No Response)

4. Have the authors made all data underlying the findings in their manuscript fully available?

Reviewer #1: (No Response)

5. Is the manuscript presented in an intelligible fashion and written in standard English?

Reviewer #1: (No Response)

6. Review Comments to the Author

Reviewer #1: This article reports the role of various adhesion forces and their proportion to the abalone adhesion force. Especially, the authors designed five types of force measuring plates to test the adhesion of abalone abdominal foot. Then, the proportion of various adhesion forces to the total adhesion force in the abalone abdominal foot was calculated and analyzed. This study quantifies the proportion of various adhesion forces to the total adhesion force of the abdominal foot.

Overall, the work in this article is new and contains very interesting results. The method used in this work can pave a new way for the design of bionic underwater adhesion devices. I recommend the publication of this article after minor corrections.

Minor points:

1. The mucus of the abalone gastropod can affect the observation of the gastropod microstructure. How to remove it?

2. Abalone is alive and the adhesion force of the abalone will suddenly increase under external stimulation. How to consider the effect of biological activity on adhesion results?

3. The authors mentioned that “the dead abalone abdominal foot does not produce vacuum adhesion force and van der Waals force” in line 282. Why does the van der Waals force disappear with the death of abalone?

4. The adhesion force survey of abalone and leech is not sufficient. See:

[1] Li et al., Hard to be killed: Load-bearing capacity of the leech Hirudo nipponia. Journal of the Mechanical Behavior of Biomedical Materials, 2018.

[2] Zhang et al., A Mechanics Study on the Self-Righting of Abalone from the Substrate. Applied Bionics and Biomechanics, 2020

5. The sentence in L14 to L15 needs revising.

6. Between the number and its unit, there should be a blank.

7. The symbol “;” is seldom used in English articles.

8. The mechanical parameters should be correctly written. If they are varibles, they should be in the italic format.

9. The symbol “Pi” in Eq. (1) is a constant, and it is not italic.

7. PLOS authors have the option to publish the peer review history of their article (what does this mean?). If published, this will include your full peer review and any attached files.

Reviewer #1: No

While revising your submission, please upload your figure files to the Preflight Analysis and Conversion Engine (PACE) digital diagnostic tool, https://pacev2.apexcovantage.com/. PACE helps ensure that figures meet PLOS requirements. To use PACE, you must first register as a user. Registration is free. Then, login and navigate to the UPLOAD tab, where you will find detailed instructions on how to use the tool. If you encounter any issues or have any questions when using PACE, please email PLOS at figures@plos.org. Please note that Supporting Information files do not need this step.<quillbot-extension-portal></quillbot-extension-portal>

---

## [Author Response · Author response to Decision Letter 1]

27 Feb 2023

1.In the pretreatment process of microscopic observation of abalone gastropod samples by SEM, the abalone gastropod sample was rinsed with phosphate buffer (PBS) for 3 times, each time for about 15min. This process can remove the mucus on the surface of the gastropod.

2.During the tensile test of abalone, the force measuring plate and abalone adhere on its surface are put on the test machine. The force measuring plate is fixed first, and then the abalone shell is hooked with the hook. The lifting speed of the tensile testing machine is 100mm/min.

When the abalone shell is hooked with the hook, the abalone is stimulated. Due to the instinct of the creature to seek advantages and avoid harm, the adhesion force between the gastropod and the force measuring plate suddenly increases. At this time, the test machine has not started. When the test machine is started, the speed of the test machine stretching upward is very slow (100mm/min), so that the abalone has enough reaction time to further improve the adhesion force between the gastropod and the force measuring plate to resist the upward tension. Therefore, the adhesion force results obtained in the tensile test have taken into account the effect of biological activity on the adhesion results.

At the same time, the biological activity of each abalone has a slight difference effects on the adhesion force results. Therefore, this study reduced the effect of the difference in adhesion force results by averaging multiple tests.

3.The van der Waals force is generated when a large amount of fibers on the surface of the abalone gastropod come into contact with the force measuring plate. When the abalone is alive, the fibers on the surface of the gastropod are also active and flexible. Therefore, when abalone is placed on the force measuring plate, the fibers distribution at the microscopic level can change with the change of the surface morphology of the force measuring plate, and the top of the fibers can effectively contact with the force measuring plate to generate van der Waals force.

When the abalone died, the surface of its gastropod contacted with the force measuring plate did not produce van der Waals force. 

The dead abalone test sample used in this experiment was obtained from the force measuring plate after the abalone died naturally on the force measuring plate. When the abalone died, the fibers on the surface of the gastropod also lost their life activity, and the fibers became hard and lost their flexibility. The fibers on the gastropod are largely lodging, making it difficult to form an upright state. Therefore, when abalone is placed on the force measuring plate, the fibers distribution at the microscopic level can not change with the change of the surface morphology of the force measuring plate, and the top of the fibers can not effectively contact with the force measuring plate to generate van der Waals force.

4.I have added the corresponding references in the text.

5.I have made corresponding modifications to the sentence in L14 to L15.

6.I have checked the full text and added a blank between the number and its unit.

7.I have checked the full text and modified the symbol “;”

8.I have checked the full text and modified the variables to italic format.

9.I have made corresponding modifications.

---

## [Editor Report · Decision Letter 2]

4 May 2023

PONE-D-22-19954R2

Abalone adhesion: the role of various adhesion forces and their proportion to total adhesion forcePLOS ONE

Dear Dr. xi,

Thank you for submitting your manuscript to PLOS ONE. After careful consideration, we feel that it has merit but does not fully meet PLOS ONE’s publication criteria as it currently stands. Therefore, we invite you to submit a revised version of the manuscript that addresses the points raised during the review process.

The authors corrected all the necessary corrections. However, in the abstract, the results are not included. There should be some quantitative outcomes. 

We look forward to receiving your revised manuscript.

Kind regards,

Arumugam Sundaramanickam, PhD

Academic Editor

PLOS ONE

Journal Requirements:

Reviewers' comments:

<quillbot-extension-portal></quillbot-extension-portal>

---

## [Author Response · Author response to Decision Letter 2]

9 May 2023

Response to Reviewers

Minor points:

The authors corrected all the necessary corrections. However, in the abstract, the results are not included. There should be some quantitative outcomes.

 I have added some quantitative outcomes in the abstract.

---

## [Editor Report · Decision Letter 3]

19 May 2023

Abalone adhesion: the role of various adhesion forces and their proportion to total adhesion force

PONE-D-22-19954R3

Dear Dr. xi,

We’re pleased to inform you that your manuscript has been judged scientifically suitable for publication and will be formally accepted for publication once it meets all outstanding technical requirements.

Kind regards,

Arumugam Sundaramanickam, PhD

Academic Editor

PLOS ONE
---

## [Editor Report · Acceptance letter]

2 Jun 2023

PONE-D-22-19954R3 

Abalone adhesion: the role of various adhesion forces and their proportion to total adhesion force 

Dear Dr. xi:

I'm pleased to inform you that your manuscript has been deemed suitable for publication in PLOS ONE. Congratulations! Your manuscript is now with our production department. 

Kind regards, 

on behalf of

Professor Arumugam Sundaramanickam 

Academic Editor

PLOS ONE